# Inborn Errors of Metabolism in the Era of Untargeted Metabolomics and Lipidomics

**DOI:** 10.3390/metabo9100242

**Published:** 2019-10-21

**Authors:** Israa T Ismail, Megan R Showalter, Oliver Fiehn

**Affiliations:** 1National Liver Institute, Menoufia University, Shebeen El Kom 55955, Egypt; israataher2015@gmail.com; 2NIH West Coast Metabolomics Center, University of California Davis, Davis, CA 95616, USA; mshowalter@ucdavis.edu

**Keywords:** mass spectrometry, LC-MS, phenylketonuria, organic aciduria, lysosomal storage disease, mitochondrial disorders, aminoacidemia

## Abstract

Inborn errors of metabolism (IEMs) are a group of inherited diseases with variable incidences. IEMs are caused by disrupting enzyme activities in specific metabolic pathways by genetic mutations, either directly or indirectly by cofactor deficiencies, causing altered levels of compounds associated with these pathways. While IEMs may present with multiple overlapping symptoms and metabolites, early and accurate diagnosis of IEMs is critical for the long-term health of affected subjects. The prevalence of IEMs differs between countries, likely because different IEM classifications and IEM screening methods are used. Currently, newborn screening programs exclusively use targeted metabolic assays that focus on limited panels of compounds for selected IEM diseases. Such targeted approaches face the problem of false negative and false positive diagnoses that could be overcome if metabolic screening adopted analyses of a broader range of analytes. Hence, we here review the prospects of using untargeted metabolomics for IEM screening. Untargeted metabolomics and lipidomics do not rely on predefined target lists and can detect as many metabolites as possible in a sample, allowing to screen for many metabolic pathways simultaneously. Examples are given for nontargeted analyses of IEMs, and prospects and limitations of different metabolomics methods are discussed. We conclude that dedicated studies are needed to compare accuracy and robustness of targeted and untargeted methods with respect to widening the scope of IEM diagnostics.

## 1. Introduction

The measurement of metabolites in the diagnosis of inborn errors of metabolism (IEMs) was established 55 years ago by Dr. Robert Guthrie for phenylketonuria (PKU) after its discovery by Dr. Asbjørn Følling three decades earlier [1]. PKU is an inborn error of phenylalanine amino acid metabolism, characterized by an increase in phenylalanine and its metabolites [2]. More than 10 years later, gas chromatography coupled to mass spectrometry (GC-MS) was used to extend the initial IEMs screenings beyond (phased-out) bacterial inhibition assay (BIA) tests, radioimmunoassays, and enzyme-immunoassays [3]. Newborn screening programs (NBS) were slowly established as part of preventive medicine. The role of NBS programs is the presumptive identification of diseases in apparently healthy subjects through application of various specific tests [4]. In 1968, screening guidelines were proposed by Wilson and Jungner and were supported by the World Health Organization [5]. In the USA, states initially dedicated NBS programs to only two to three diseases, later gradually increasing the number of tested diseases [6,7]. In the 1980s, IEMs were screened by gas chromatography/mass spectrometry (GC/MS). Yet, in the late 1990s, the advent of liquid chromatography/mass spectrometry (LC-MS) enabled the rapid diagnosis of 22 IEM diseases in parallel [3]. LC-MS/MS based methods target a suite of critical metabolites and represent the most widely used metabolomic assays, implemented in clinical routines worldwide. These methods reduce the demand for previously used, time-consuming or less accurate measurements [8]. Today, LC-MS and direct infusion tandem mass spectrometry (MS/MS) are considered the gold standard for measurements of inborn errors in metabolism. Tandem mass spectrometry screening of IEMs uses commercially available kits to detect several common disorders with a single injection [9,10]. MS/MS assays expanded the screening list of IEM diseases from the most common disease categories, aminoacidemia, organic aciduria, urea cycle disorders [11], and galactosemia [12], to include less common diseases. IEM screening methods now include fatty acid oxidation defects [13], purines and pyrimidines disorders [14] and other diseases [15,16,17,18,19]. Stable-isotope-labeled reference compounds are now routinely used as internal standards. Metabolic profiling assays have allowed the study of different phenotypes of the same disease, as it was used for diagnosis of Mucopolysaccharidoses (Lysosomal Storage Disease) types II, IVA, and VI [20]. This progress has allowed for better disease diagnosis and understanding of IEM categories.

In parallel to measurements by mass spectrometry, genetic screening of IEMs started in the 1990s [21]. Today, next generation sequencing enables genetic screenings as secondary diagnostic tool [22]. Sequencing based techniques have drawbacks such as cost, delayed results, detection of variants with uncertain clinical significance [23], and ethical problems [24]. These problems limit its use in most diagnostic algorithms [23]. Yet, advances in genomics also led to an uneven acceptance of the Wilson and Jungner guidelines [25]. Genetic variants with uncertain significance may induce incidental findings outside IEMs [23]. Nonvalidated genetic variances caused by variable penetrance or random X-chromosome activation affect subsequent adaptation of gene assays for IEM diagnosis [26,27]. In effect, testing for genetic variants in IEMs might best be combined with other omics data, including metabolomics, to better understand the mechanisms how complex phenotypes are associated with specific primary mutations [28].

Nowadays, targeted metabolite assays represent the key technology in NBS worldwide [29]. However, there is a high degree of heterogeneity and a lack of consensus in the tested diseases among various NBS programs. Within Europe, the number of screened IEM diseases in NBS programs is noticeably different. Finland screens only for congenital hypothyroidism, whereas Austria screens for of 29 diseases [30]. Even within a country, such as Belgium, the number of screened diseases varies in different regions and states [29,30], similar to the situation in the USA. This discrepancy in adopting IEM screening methods also appears in the policies of cost reimbursements [31] or the thresholds of biomarkers used for screening the same disorder [29,32]. The need to harmonize NBS is challenged by the lack of sufficient information about various phenotypes, prevalence and natural history of the diseases [33]. Most current research focuses on a single disorder or a group of linked disorders directed to a specific group or populations [34,35,36,37,38]. The threshold for reporting false diagnosis also hampers direct comparisons of NBS results across nations [33,39].

The use of untargeted metabolomics could complement current targeted metabolite assays. Untargeted metabolomics has the prospect of studying a wider range of metabolic pathways and to provide a broader view of the true metabolic phenotype of diseases [40]. Untargeted metabolomics methods can be merged with aspects of classic targeted assays by validating methods for specific known IEM biomarkers, including the use of stable-isotope labeled internal standards, while simultaneously broaden the scope of analyses to semiquantified metabolites by accurate mass profiling [41]. Untargeted metabolomics can deepen the understanding of disease pathways and support new discoveries that may open up new treatment options [19]. This premise of untargeted metabolomics was exemplified already 12 years ago, detailing the effects of a single altered gene on multiple biochemical pathways for two IEM diseases [42]. However, it is yet unclear how untargeted metabolomics will fulfill clinical requirements with respect to costs, speed, accuracy, and repeatability.

In this review, we will discuss IEM diseases with emphasis on screening and diagnosis using targeted versus untargeted mass spectrometry (MS) approaches. We will give an overview of studies, including application of matrices analyzed, instrumentation, and data processing methods. We will also discuss challenges and obstacles to adapt untargeted mass spectrometry in IEMs screening and diagnosis.

## 2. Overview of Inborn Errors of Metabolism Diseases

IEM diseases are a group of inherited genetic diseases resulting from total or partial absence or deficient activity of an individual enzyme, structural protein, or transporter molecule [36]. Recent articles reported more than 1015 known IEMs [43]. There is a lot of discrepancy in the literature estimating the overall incidence, but recently estimated incidence rates are one in 800–2500 live births [22,44]. Individual disorders are more frequently diagnosed today, but such diseases are still uncommon and vary in different countries and regions [45].

A recent attempt for classification associated with the prevalence of each category of IEMs has been reported by Ferreira and his colleagues, (Figure 1). They used specific solid criteria to establish a classification and prevalence of IEMs with the intent to form the first formal nosology of IEMs and to update this categorization regularly [43].

Studies from different countries have found wide disparities in the reported incidence rates of IEM diseases (Figure 2)**.** These discrepancies of IEM categories appear in overall incidence but also for specific categories including organic acid disorders, lysosomal storage disorders, fatty acid metabolism disorders, mitochondrial disorders, urea cycle disorders, amino acid metabolism disorders, and carbohydrate metabolism disorders.

However, there is a lack of data regarding the causes of disparities in overall incidence and rates of individual disorders worldwide. Higher rates of consanguineous marriage in some countries or regions can increase the incidence rates up to 50-fold [46]. Regional differences in genetic diversity, high rates of inbreeding and large family size [47] all contribute to variable incidences of the overall incidence of IEMs. Further confounding the understanding of overall incident rates of IEMs is the variation of study methods and duration used for the same disease in different countries [48].

## 3. Inheritance and Causes

Most IEMs are inherited in an autosomal recessive (AR) manner [49]. In AR diseases, phenotypes will only manifest if both parental copies of the mutated allele are inherited. Homozygote gene mutation inheritance can occur by either consanguineous marriage or by a random mutation in the second allele in heterozygote parents. Some IEMs are inherited as X-linked alleles, and therefore have higher incidence rates in males. Due to variable random X-chromosome inactivation syndrome in females, X-linked IEMs diseases can have highly variable manifestations from one tissue to another and from one female (if so affected) to another. A minority of IEMs are inherited in an autosomal dominant (AD) pattern. Another rare mode of inheritance is IEMs linked to mitochondrial DNA only of maternal origin, as in subsets of respiratory chain disorders [50,51]. The pathophysiology behind most IEM disorders is a specific enzyme defect that results in an inadequate conversion of substrates into their direct products. That defect leads to accumulation of upstream substances which induce toxic effects and abnormal alternative substrate metabolism, in addition to reduced downstream essential products [52] (Figure 3).

## 4. Classifications

IEM disorders are often characterized by alteration of multiple overlapping metabolic pathways. Different classifications have emerged to categorize IEM diseases to enable easier clinical and laboratory diagnosis and facilitate treatment [53] such as pathophysiological classification of IEMs, (Figure 4). IEMs were assorted as disorders of carbohydrate metabolism, disorders of amino acid metabolism, disorders of organic acid metabolism, and lysosomal storage diseases [54]. However, more complex classification systems have recently been proposed [55].

The Society for the Study of Inborn Errors of Metabolism (SSIEM) considers clinical aspects of IEMs in a specific classification. The SSIEM classification assorts large numbers of individual disorders according to their biochemical pathways and common pathophysiological mechanisms [4,44]. Metabolomics assays must be developed to reflect such classifications.

## 5. Clinical Presentation and Outcomes

IEMs may present at various ages in different ways. Clinical presentation of the disease can occur even before birth, at birth, or during the first days of life as deterioration after normal birth and delivery [56]. Errors in fetal metabolism may be associated with developing maternal complications during pregnancy such as fatty liver and HELLP (Hemolysis, Elevated Liver Enzymes, Low Platelets) syndrome [34]. At birth, inborn errors can manifest as perinatal asphyxia, or later as nonspecific chronic manifestations such as delays in childhood developmental milestones. Acute metabolic decompensation in the neonatal period may also present as severe acidosis, alkalosis, or hypoglycemia [15,48,56]. Most IEM babies born at term seem to be well, but then deteriorate quickly, even when babies do not receive oral feeds.

This phenomenon is caused by changes in catabolism which occurs normally in the first days of life, leading to an accumulation of metabolites that cause toxic manifestations. The rate of deterioration is variable according to disease type, the extent of the disease and the most affected organ. For example, inborn errors in metabolism may appear as neurological symptoms, disorders of acid-base balance, unexplained hypoglycemia, cardiomyopathy, hepatic deterioration, or sudden death. Other diseases have more subtle presentations, such as a characteristic odor which is not commonly detected [56,57]. In general, IEMs can be pleiotropic (affecting more than one system or organ) or have a localized effect [58]. IEM diseases are responsible for a significant portion of childhood disability and deaths [43].

## 6. Diagnosis and Screening Inborn Errors of Metabolism

Clinical manifestations of IEMs are overlapping. Therefore, clinical approaches to diagnosing IEMs are very difficult, especially for rarer disease variants. Presenting symptoms in infants can have a wide variety of possible causes. Distinctive facial abnormalities are linked to a group of diseases such as lysosomal storage disorders, pyruvate dehydrogenase deficiency, glutaric aciduria type II, cholesterol biosynthesis defect, glycolysalation disorders, and disorders of peroxisomal biogenesis [59]. Infants born with cardiac manifestation should be investigated for mitochondrial respiratory chain defect, fatty acid oxidation disorder, and Pompe’s disease [60]. Congenital disorders of glycosylation disorders could present with dysmorphic features and/or cardiomyopathy [61]. Hypoglycemia is a critical manifestation of carbohydrate, fat metabolism disorders, and any IEMs with direct or indirect hepatic insult [62]. Unexplained persistent metabolic acidosis in infants is a finding associated with poor outcome due to wide IEM disorders related either directly or indirectly (multi organ failure) [60,63]. Acute encephalopathy and other neurological manifestations in infants have more than 50 causes of various IEM diseases. These diseases represent organic aciduria, urea cycle disorders, mitochondrial, lysosomal, and peroxisomal disorders [64]. Infants with IEM presenting with liver dysfunctions have differential diagnoses ranging from galactosemia to lysosomal storage disorders. Liver manifestations also carry the possibility of nonmetabolic diseases as sepsis and hypopituitarism [65]. It is therefore impossible to use a universal clinical protocol for all IEMs, making it very difficult to limit differential diagnosis only based on clinical findings. Clinical data are not sufficient for final diagnoses. Instead, laboratory data have become the primary clue for diagnosis [66], [67]. Romão et al. reported that from 144 clinically diagnosed infants with IEM diseases, only 12 infants had confirmed diagnoses using laboratory investigation [67].

Diagnosis of IEMs is based on biochemical tests that are divided into two approaches: (1) screening tests to detect possible abnormal levels of metabolic biomarkers in blood or urine before the disease manifests and (2) tests to detect specific pathognomonic biomarkers [68]. All biochemical diagnostics procedures rely on the identification of abnormally high levels of the main substrates or of byproducts that arise from alternative pathways upstream of the enzymatic blockage. These compounds can be detected together with lower levels of the product of that enzyme or any of its downstream metabolites [52]. Metabolic investigations are the primary information regarding diagnosis [69,70] and are essential for effective treatment monitoring that could be curative, prevent continued disease progression, or limit disability [71].

Screening is an important tool for primary disease prevention. In contrast to diagnostic investigations, the screening tests aim to detect diseases in the latent asymptomatic stage of the disease to facilitate intervention and improve outcomes [72]. IEM screening tests should be directed to all newborns with accepted reliability, cost, and validity [5]. NBS programs had a dramatic effect on improving the outcomes in many IEMs. IEMs can be detected at an asymptomatic stage, enabling rapid medical interventions that positively change the progression of the disease [52] to prevent life-threatening or long-term sequelae [73]. The impact of early screening was reported in various diseases as an improved neurocognitive outcome in early diagnosis of phenylketonuria and congenital hypothyroidism [32,74]. NBS improves the mortality in cases of medium-chain acyl-CoA dehydrogenase (MCAD) deficiency [75], the overall outcome in cystic fibrosis (CF) [76], and primary immune deficiencies [77]. Yet, current NBS programs that rely on targeting few specific metabolites still produce erroneous results, especially in stressed infants who are born prematurely or who present with low birth weight [78].

In the symptomatic stage of the disease, severity and onset of symptoms influence the investigations and diagnostic methods used. However, as many IEMs present with very similar symptoms, it is necessary to perform multiple diagnostic investigations concurrently [60]. The investigations of clinically manifested IEMs include tests to detect the nature and degree of system affection, the type and level of toxic substances, in addition to generalized metabolite screening tests. This is often performed through targeted mass spectrometry based metabolite assays [52,60,79].

## 7. Metabolomics Technologies Used in Inborn Errors of Metabolism

Metabolomics seeks to measure the complement of endogenous and exogenous small molecules in a given sample. Metabolomics analysis in cells, tissues, and body fluids reflects the biochemical status of the sample a direct readout of the current phenotype (normal or pathological) of the living organism [80,81]. These compounds include molecules participating in catabolic and anabolic pathways, small molecule regulators, epimetabolites [82], environmental factors [28], and microbial metabolites [83]. While different molecular techniques can identify changes at the genetic level, it is not always clear how either mutations, single nucleotide polymorphisms, or other changes to DNA or gene expression will be translated in the cell. Metabolism receives inputs from every level of cellular regulation and better represents the true cellular phenotype. In this regard, metabolomics has been used successfully to give insights about the cause of diseases, discovered novel biomarkers, and shed light on the impact of drug metabolism and drug effects in vivo [84,85,86].

Metabolomics analysis in clinical medicine is most commonly performed using nuclear magnetic resonance (NMR) spectroscopy and mass spectrometry (MS) [85]. NMR spectroscopy measures the magnetic property of atomic nuclei [87]. NMR spectroscopy has been described as a fast, reproducible, and non-invasive method [88,89]. However, NMR has become less widely used in studies for newborn screening in blood. While NMR may give stereochemical and structural insight into analysis of isolated compounds, in complex mixtures of compounds (such as in blood), it measures far fewer metabolites per sample than mass spectrometry. NMR is less suited for the measurement of low abundant compounds because it is considerably less sensitive than mass spectrometry [90].

Mass spectrometry is used in two different modes in IEMs screening: either by direct injection of the sample to the ionization source of the mass spectrometer or by separation mixtures of compounds by gas chromatography (GC) or liquid chromatography (LC) prior to MS detection. GC or LC methods reduce the complexity of chemical mixtures prior to mass spectrometry analysis [86] and is well suited to separate isomers. Hence, GC-MS and LC-MS are better suited to detect and quantify compounds in metabolomics analyses than NMR or direct infusion MS. Chromatographic separation adds additional information about the metabolites and is used to aid in compound identification [86,91]. GC-MS is optimal for the analysis of volatile metabolites, but it can also be used for analysis of primary metabolites when chemical derivatization schemes are used. LC-MS is the most common method used for both polar and nonpolar compounds [92].

The introduction of ultrahigh-performance liquid chromatography (UHPLC) and tandem mass spectrometry (MS/MS) have improved sample throughput and analytical sensitivity of a wide range of metabolites [93,94,95]. Recent advances in high resolution mass spectrometry (HRMS) yields better mass accuracy and sensitivity than classic nominal mass measurements. HRMS enables detecting low abundant compounds at less than ng/L [96]. In profiling mode, HRMS can increase the number of detected metabolic signals to thousands of features even in small sample sizes, like dried blood spot (DBS) samples [97]. In metabolomics newborn screening programs of IEMs, tandem mass spectrometry (MS/MS) is the principle approach because of its rapid turnover, high specificity to detect target metabolites, high sensitivity, and low sample volume requirements [44,98].

## 8. Matrix

Metabolomics in biomedical studies generally uses biofluids, cells, and tissues as the primary matrix to generate metabolic signatures. In clinical practice, urine, serum, and plasma are easy to collect and to prepare. They present the most commonly used biofluids [99,100,101]. Currently, the most commonly used sample types in newborn screening programs worldwide is blood in the form of DBS. Dried blood spots have small sample volumes, are easily transported, are less biohazardous, and are not as invasive to collect compared to plasma [102,103]. The use of DBS for the screening of IEMs was reported by Guthrie and his colleagues [1] for screening PKU, orotic aciduria [104], and aminoacidopathies [105]. The use of DBS and other dried biofluids have been validated in multiple studies [106,107] as a good alternative to liquid samples for metabolomics owing to their low volume and cost with easy handling [108,109]. Urine is used also for screening, as it is easy to collect, especially using filter paper. However, the risk of contamination by feces in early age children limits collection quality, and urine cannot detect all diseases associated with metabolites, limiting its usage [78].

In addition, other biosamples could be used, for example cerebrospinal fluid (CSF) [110] and saliva [111]. These matrices are less commonly used due to difficulties with clean sample collection or ease of sample collection. They are only used only for the diagnosis of specialized disorders. For example, CSF is most commonly used for screening monoamine neurotransmitter deficiencies [112]. Cord blood has been used in the detection of thyroid stimulating hormone (TSH) in the screening of congenital hypothyroidism [113]. Human skin fibroblasts were used for in vitro detection of the defect in isoleucine catabolism in ethylmalonic encephalopathy by Sahebekhtiari et al. [114]. Ethylmalonic encephalopathy is an organic aciduric disorder characterized by developmental delay, hypotonic manifestation, and excretion of ethylmalonic acid (EMA) in urine [114,115]. While most studies utilize targeted MS approaches, untargeted studies were conducted for comparison of metabolomics profiles in different matrices. Koulman and Kennedy used untargeted lipidomics to compare DBS, plasma, and whole blood profiles [116]. Similarly, metabolic profiles in cerebrospinal fluids, plasma, and urine were compared [116].

## 9. Methods of Metabolomics Analyses

Metabolomics is divided into two approaches: studies looking only at specific compounds (targeted metabolomics) and studies that aim at measuring all compounds (untargeted metabolomics), Table 1. Targeted metabolomics methods detect and quantify a group of compounds using internal standards and may be compared to a known reference range [117]. Only compounds established as part of the method will be measured, while other compounds in the sample are not detected. The most commonly used clinical approaches seek to accurately quantify all metabolites of interest while not collecting data on the remaining small molecules present [41]. In untargeted metabolomics assays, samples are analyzed to detect as many compounds as possible. Untargeted metabolomics therefore detects both known and unknown compounds. The measurement of compounds outside of the diagnostic set can provide deeper understanding of the disease [118]. Using internal standards with untargeted metabolomics might achieve the same quantification confidence as in targeted metabolomics [41].

## 10. Targeted Metabolomics in the Screening and Diagnosis of Inborn Errors in Metabolism

Traditionally, the study and diagnosis of IEMs is performed using a panel of targeted analyses with dedicated analytical protocols. These protocols cover a selected panel of diseases by quantifying the metabolites in a disease pathway and metabolite levels are then compared to the range of healthy (normal) metabolic concentrations [66,119,120]. For some diseases, metabolite measurements are assembled into panels [121,122]. The most commonly screened panels of IEMs are using markers of amino acids, fatty acid oxidation, and organic acid metabolism disorders [123,124,125]. While targeted mass spectrometry methods are often used in the clinic and provide vital diagnostic information, they fail to measure the diverse range of compounds found in biofluids.

Despite its unquestionable role, current target-based newborn screening programs yield only a snapshot of all metabolic alterations. Many IEMs cannot be identified by current routine targeted metabolite analyses [44]. Even when targeted screening protocols focused on a single disease or a group of related diseases, up to 79% false positive results were reported in a study in Taiwan [126]. In 2005, newborn screening error rates were estimated to occur in 2500 to 51,000 cases in the United States, per expected specificities of individual metabolite target tests [127]. Such false positive results have great psychological impact on parents [128]. In addition, there are added costs of secondary confirmatory diagnostic testing, follow-up, and primary medical management until the correct diagnosis is revealed [129].

False interpretation of disease manifestations by the parents or family doctors as well as incomplete description of patient symptoms by the parents may lead to inappropriate diagnostic panels. Hence, there is a high risk that improper tests are performed by metabolite target screening methods, increasing the rate of false negative results and delaying accurate diagnoses [118,124]. Using different cut offs for the same disease is responsible for missed cases as well as for false positive results [124,130]. Both false negative and false positive results in newborn screening have many causes. A single biomarker or set of two metabolites can act as biomarker for more than one IEM disease. For example, methionine and cysteine levels are used for the diagnosis of homocystinuria, methionine adenosyltransferase deficiency, and adenosylhomocysteine hydrolase deficiency. Histidine levels are used for the diagnosis of histidinemia and formiminotransferase deficiency [131]. Further complicating correct diagnoses is the fact that many amino acid biomarkers may be affected by other factors such as feeding prior to screening or day time of screening [124,132].

The economic evaluation of targeted newborn screening programs has been studied in different areas over the last decade. Thiboonboon et al. described extended metabolic screening as not cost-effective in Thailand [133]. In the UK, newborn screening is recommend only as cost-effective for a limited subset of diseases, dedicated for PKU and Medium Chain Acyl COA Dehydrogenase Deficiency (MCAD), excluding many other inherited metabolic diseases [134]. For the US state of Texas, it was reported that expanded newborn screening increases the costs to the payer [135]. Newborn screening programs are still not used in all countries worldwide, especially in developing countries [120]. The American College of Medical Genetics have recommended a uniform panel of conditions formed of 29 diseases together with 25 related diseases, including outcomes and guidelines [136]. However, these recommendations are given for a specific country but are not used worldwide. Even within the US, different states adopted varying parts of these recommendations and did not implement single standardized sets [29].

In addition to limits of testing policies, targeted metabolomic analysis in classic newborn screening programs limits the discovery of novel metabolic defects due to its focus on a single panel of known metabolic pathways. Providing a broader understanding of the pathophysiology and the metabolic interactions behind the phenotype of the disease is important to design better treatment regimens [118,137]. Current newborn screening programs are only performed for treatable diseases. This practice is in alignment with the Wilson and Jungner principles published 42 years ago [5]. Since then, genomics tools, and now metabolomics, have expanded the possibility to characterize previously undiagnosed diseases. The World Health Organization noted an uneven adoption of the Wilson and Jungner principles in the era of genomic testing [25]. In our opinion, patients have an ethical right to be diagnosed, even if there are no validated treatment options available. Genomic and metabolomic testing will detail mechanisms of IEM etiologies, provide a baseline of prevalence of subcategories of IEMs, and possibly provide a basis for developing treatments as outlined in a recent white paper on the use of metabolomics in precision medicine [138]. Treatment options have shown rapid advances in last decades [139,140,141]. Yet, neither genomic nor metabolomics technologies are mature enough to justify mass IEM screening programs at this point. Cost-benefit analyses may show that current targeted programs still underserve populations in need. Metabolomic and genomic databases must be constructed and validated on population scale to prevent over diagnosis of IEMs. Yet, technologies have matured sufficiently to the point that genomic and metabolomic studies can now be conducted to validate the omics premises in the context of IEMs. 

With recent characterizations of novel IEMs, researchers are beginning to recognize the drawbacks of targeted metabolite methods in newborn screening program, especially for diseases that are difficult to diagnose. Lysosomal storage diseases have a high degree of phenotypic and genetic variability and are characterized by multisystem effects [142]. Fabry disease, a lysosomal storage disorder caused by a deficiency of the α-galactosidase-A enzyme and characterized by an accumulation of globotriaosylsphingosine (lyso-Gb_3_) and globotriaosylceramide (Gb_3_), has proven challenging to be correctly diagnosed. Auray-Blais et al. analyzed plasma and urine samples of Fabry disease patients using LC-time of flight mass spectrometry and found that more than 20 isoforms of Gb_3_ were responsible for the disease severity and prognosis. Such detailed results were previously missed by targeted approaches [143,144,145,146]. Thus, the need for broader techniques for metabolite screening should be considered to correctly classify the growing numbers of IEMs. Untargeted metabolomics using high resolution mass spectrometry might be best suited [118].

## 11. Untargeted Metabolomics in the Screening and Diagnosis of Inborn Errors of Metabolism

In contrast to targeted metabolomics approaches, untargeted metabolomics aims to measure all detectable analytes in a sample, including unidentified metabolites [41]. Untargeted metabolomics is the most frequently used technique to elucidate the pathophysiological background and detect novel biomarkers in a broad range of diseases [147,148,149,150]. A forerunner of untargeted metabolomics was the use of gas chromatography-mass spectrometry (GC-MS) for urinary organic acids analysis and IEM diagnosis [151], used since the 1970s. High levels of urinary organic acid characterize organic aciduria as IEM [152]. GC-MS analysis of urinary organic acid is now presented in qualitative or semi quantitative manner by adding specific metabolites standards [153,154].

Sample preparation for untargeted metabolomics aims at broad scale metabolome coverage without discriminating against specific classes of small metabolites while still removing proteins and other macromolecules [155]. Typically, extractions are performed using liquid-liquid extraction or, less commonly, solid-phase extraction. Three crucial parameters define sample preparation for untargeted metabolomics platforms: repeatability (precision), metabolome coverage, and the extent of protein precipitation. The most often used method for protein depletion in biological samples is cold precipitation by organic solvents, then centrifugation [156,157]. Subsequently, the complex chemical mixtures of biosample extracts are separated by liquid or gas chromatography columns. Hence, retention times, best specified in relation to internal standards, must be given in IEM reports to enable independent verification of results, along with information on compound (mass-to-charge ratios, *m/z*) and MS/MS fragmentation spectra. Chromatographic retention times represent the degree of interaction of chemicals with the column adsorbent material, while accurate mass *m/z* and MS/MS spectral data can be compared to authentic chemical standards to verify claims on compound identification in IEM reports [158,159].

For untargeted assays, high resolution mass spectrometers provide the best sensitivity, specificity and coverage. Quadrupole time-of-flight MS (QTOF MS or TTOF MS) and quadrupole Orbitrap MS (Q-Exactive) instruments have full scan modes and ability to fragment ions to provide a panoramic view of metabolites in biological samples. Triple quadrupole MS (QQQ MS) are most useful for targeted metabolite newborn screening, especially for absolute quantifications [160]. Yet, on full-scan mode, QQQ are less sensitive than QTOF or Q-Exactive MS instruments. QQQ also do not yield accurate masses and are therefore unsuitable to identify novel biomarkers.

Untargeted metabolomics produces large and chemically diverse datasets. Such datasets comprise signals of both of known and unknown chemical structures and require careful interpretations. Statistical analyses of large datasets must account for false discovery rates to adjust for multiple testing. In addition to univariate analysis, data can also be processed by multivariate data analysis, including enrichment statistics [161]. Statistical analysis associates the most important metabolites to mechanisms of diseases, diagnosis, treatment and prognosis [97,162,163], yet, such conclusions must always be validated by independent secondary studies. 

Many clinical and biomedical studies have reported using global untargeted metabolomics [164,165], Table 2. Untargeted metabolomics aids better understanding of differential use of metabolic pathways that are associated with health-related phenotypes [70,166]. Untargeted metabolomics may overcome problems in targeted methods for newborn screening by increasing the number of screened IEM diseases and decreasing the incidence of false negative results [167]. Indeed, a range of IEM studies have used untargeted metabolomics with a high degree of success (Table 2). Denes et al. tested using dried blood spots by high resolution mass spectrometry on 66 samples from nine different IEMs diseases (Phenylketonuria (PKU), Medium Chain Acyl COA Dehydrogenase Deficiency (MCADD), Homocystinuria (HCY), CLD, Maple Serum Urine Disease (MSUD), Isovaleric acidemia (IVA), Propionic Acidemia (PA), and 3-MCC, Tyrosinemia, Citrullinemia and Galactosemia), in comparison to 500 control samples. With this excellent number of controls, the clear discrimination between IEM diseased patient samples and control cases appears to be primed for subsequent validation studies [108]. Similarly, Miller et al. recognized novel biomarkers and pathways of 21 IEM disorders. He analyzed 120 plasma samples of diagnosed subjects compared to 70 control samples using targeted and untargeted techniques by GC and LC-MS [168].

Untargeted metabolomics widens the range of metabolites associated with IEMs and discovers new compounds that could be potential biomarkers. Using untargeted metabolomics in phenylketonuria of patient plasma led to the identification of two new biomarkers, glutamyl-glutamyl-phenylalanine, and phenylalanine-hexose [177]. These new markers showed a high degree of variation between PKU patients and did not correlate with phenylalanine levels, illustrating their potential to highlight new mechanisms of the disease that would require further validation. In a distinct study on methylmalonic acidemia and propionic aciduria plasma samples, C18 LC-TOF MS-based untargeted metabolomics was utilized. Apart from finding the known biomarker propionyl carnitine and other acylcarnitines (such as isovaleryl carnitine), γ-butyrobetaine showed significant differences among the two diseased groups in comparison to control samples [42].

## 12. Lipidomic Studies in Inborn Errors of Metabolism

Metabolites can be divided per their physicochemical properties such as water-soluble (hydrophilic) and lipid-soluble (lipophilic) molecules. Hydrophilic molecules are the domain of primary metabolism such as sugars, amino acids, organic acids, or nucleotides. Screening lipophilic compounds has been termed lipidomics [178]. It routinely distinguishes lipid species such as ceramides, sphingomyelins, and phospholipids [179]. Such polar lipids are the main constituent of cell membranes, myelin sheaths, and intracellular organelle structures. Oxidized eicosanoid lipids and phosphatidylinositol-lipids act in inter- and intracellular signaling, while neutral lipids are the main reservoir for energy [180]. Hence, dysfunction in lipid metabolism results in various metabolic diseases including metabolic syndrome [106]. More than 100 different IEMs have been associated with abnormal lipid metabolism [180] such as peroxisomal disorders [181], fatty acid oxidation defects [56], and cholesterol biosynthesis pathways disorders [182]. Lipidomics can be used in targeted or untargeted methods to find biomarkers for diagnosis and understanding of the pathophysiology of lipid-related diseases [181]. Lipidomics extractions are usually performed by liquid-liquid extraction, followed by LC-MS analyses [178,183].

Acylcarnitines are important classical targets for diagnosis of IEMs such as organic aciduria, mitochondrial and fatty acid oxidation defects [184]. Few studies have used untargeted lipidomics for screening and diagnosis of IEMs. Plasma lipids were assessed in two different fatty acid oxidation disorders (LCHAD and CPT2), discovering altered partitioning of long-chain fatty acids into complex lipids [173]. Plasma and urinary lipids were studied in a cohort of Fabry disease patients under enzyme replacement therapy, reporting increases in both sphingolipids and phospholipids [185]. A new method for lipidomics analysis from dried blood spots and LC-high resolution mass spectrometry was presented [106]. Similarly, an extensive database was built to catalogue the human lipidome of cerebrospinal fluids by LC-high resolution mass spectrometry to start investigations into biomarkers for neurological disorders [186].

## 13. Processing Raw Untargeted Metabolomics Data

With current advances in analytic methods, thousands of peaks and metabolites with good sensitivity are revealed using different platforms. The need to identify these peaks (especially those with significant effects in IEM studies) is a grand challenge for untargeted metabolomics researchers [187]. The first step in raw data processing is to produce a list of mass/retention time signals. Commonly used software packages are MS-DIAL [188], XCMS [189], MZmine2 [190], and MAVEN [191], but other tools, including licensed software, are used as well. Some software such as MS-DIAL has built-in capabilities to utilize MS/MS spectra and retention time data to identify these metabolites by mass spectral libraries [192]. The largest freely available MS/MS library is MassBank of North America [193] with over 130,000 experimental spectra and 490,000 in silico predicted spectra for lipids (LipidBlast [194]). MassBank of North America contains spectra from other open source repositories, including the Human Metabolome Database (HMDB) [187,195,196]. Apart from these open-access libraries, fee-based MS/MS repositories complement the informatics tools for compound identification, including the well-curated NIST17 database [197] and the METLIN library [197,198]. Yet, novel IEM biomarkers discovered by untargeted metabolomics or lipidomics might not be covered in common MS/MS libraries. Hence, researchers have extended chemical structure databases to include enzyme-extensions of metabolism, such as the MINE database [199] and My Compound ID that uses an evidence-based metabolome library (EML) [200]. However, matching metabolite spectra against such huge in silico databases can give rise to false positive identifications. In silico generated mass spectra are often not based on specific MS instruments or instrument parameter settings [201], and hence, in silico MS/MS spectra must be used with great caution for identifying IEM biomarkers.

## 14. IEM Screening and Diagnosis Comparing Targeted Versus Untargeted Metabolomics

A number of studies compared results between targeted and untargeted techniques [202,203]. A recent study on plasma of urea cycle disorders (UCD) identified novel altered compounds in partial ornithine transcarbamylase (OTC) deficiency disease (X linked UCD) in female participants [176]. In addition to previously known biomarkers of UCDs, this study detected metabolites associated with long term complications of UCDs and discovered off-target effects of medication [176]. However, the authors reported a long turnaround time as one of the limitations facing untargeted metabolomics, due to the lack of automated data processing. Yet, they noted that the specificity of the assay was high because several metabolites were detected only in diseased cases but not at normal baseline levels [176].

Conversely, classic NBS using targeted metabolomic approaches did not show much improvement over the past years with respect to the magnitude of false positive results. Two distinct studies reported one false positive case per every 50–300 true positive cases in the USA [204,205]. The main cause for false positive results is the low cut off values in IEM screening tests to avoid missed cases (false negatives) [206]. By and large, targeted amino acid measurements correlated to untargeted metabolomic analysis in a study on IEMs in urine and CSF [207]; however, the authors noted that tryptophan was degraded by acidification in the targeted assay but correctly quantified in untargeted metabolomics [207]. Despite being recognized as generally reliable and quantitative, targeted metabolomics may provide uncertain results in some instances, for example, for patients in borderline categories [208]. Untargeted metabolomics proved superior by detecting and analyzing key differences between aromatic amino acid decarboxylase deficiency from drug-induced metabolite elevations, specifically by detecting compounds such as dopamine 3-*O*-sulfate, vanillylmandelate, and 3-methoxytyramine sulfate that are not screened in targeted assays [209].

If targeted metabolomics yields inconclusive diagnosis results, additional tests are required such as analysis of further targets, DNA analysis, or more invasive techniques such as individual enzyme assay in cells. Such analyses require additional costs and time to achieve the accurate diagnosis [210]. These problems may be overcome by using untargeted metabolomics assays. In a study on using globotriaosylceramide (Gb3) and globotriaosylsphingosine (lyso-Gb3) as target biomarkers in urine and plasma of Fabry, it was noted that untargeted urine metabolomics revealed seven novel urinary lyso-Gb3-relateds isoforms in both male and female patients with Fabry disease, but not in healthy controls [144,145,211,212]. Hence, in this case, untargeted metabolomics yielded no false positive cases and 100% specificity, in addition to improved diagnosis for female cases which are challenging in Fabry disease diagnosis [144,145,211,212].

Screening of IEM diseases using untargeted metabolomics gives a broad overview on aberrant pathways. A large screen of multiple metabolic pathways provides more diagnostic confidence, (Figure 5). For example, Li et al. published about false diagnosis reports of arginase deficiency by an increased level of amino acid arginine obtained by targeted metabolite screening, which led to inappropriate treatment for more than four years [213]. In contrast, untargeted metabolomics study of arginase deficiency revealed alterations of more than 30 metabolic pathways linked by guanidino compounds [176], giving a higher likelihood of correct diagnostic reports.

## 15. Conclusions

Targeted metabolomics in screening and diagnosis of IEMs give a narrow view of the diseases assayed. We need to consider untargeted metabolomics as a new tool to improve the scope of IEM disease categories associated with pathophysiology, early symptoms, therapy options and follow-up strategies (Figure 6). Untargeted metabolomics will still need to remove some barriers (such as in standardization, quantification, and compound identification) to become more useful for clinicians, IEM researchers as well as the metabolomics community. To improve standardization of IEM metabotyping, three steps are needed: (1) gathering distributed data of IEM diseases to a worldwide accessible database, (2) providing open access reference databases to interpret analytical findings from different instrumentations and matrices, and (3) generalizing screening programs across countries by introducing affordable untargeted metabolomics. These actions will provide a better mechanistic understanding, disease prevention, management, and disease outcome of IEMs.

## Figures and Tables

**Figure 1 metabolites-09-00242-f001:**
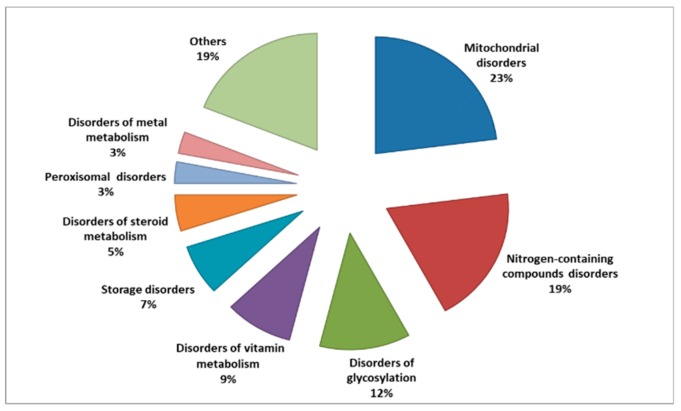
Percentage of each category of inborn errors of metabolism (IEMs) according to inclusion criteria reported by [43].

**Figure 2 metabolites-09-00242-f002:**
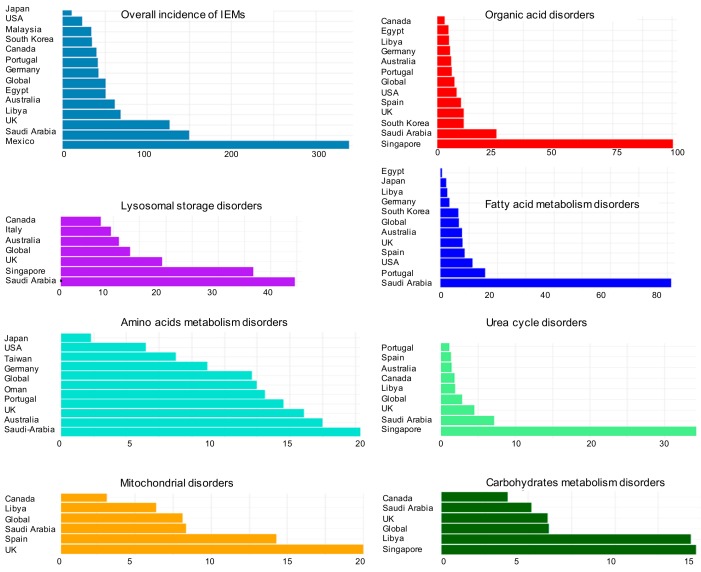
The prevalence of IEMs diseases globally and among different countries. Numbers on x-axes are given per 100,000 live births (Appendix A).

**Figure 3 metabolites-09-00242-f003:**
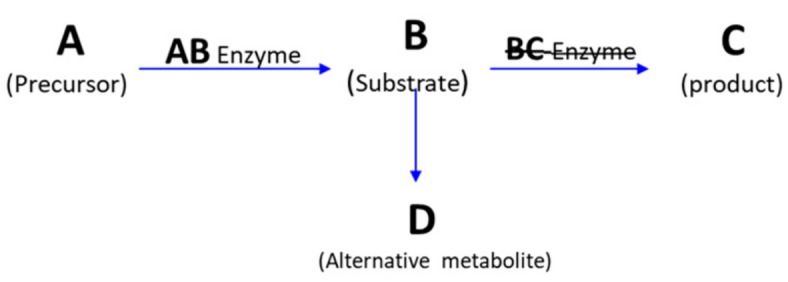
Mechanism of IEM diseases. Enzyme AB converts metabolite A to product B. A defect in enzyme BC leads to an accumulation of metabolite B that may cause an activation of the pathway BD. This causes an abnormal concentration of metabolite D and a deficiency in the end product C. Alterations in these compounds are called the metabolic signature of the “ABCD disease” which can be used for diagnosis if all compounds are detected.

**Figure 4 metabolites-09-00242-f004:**
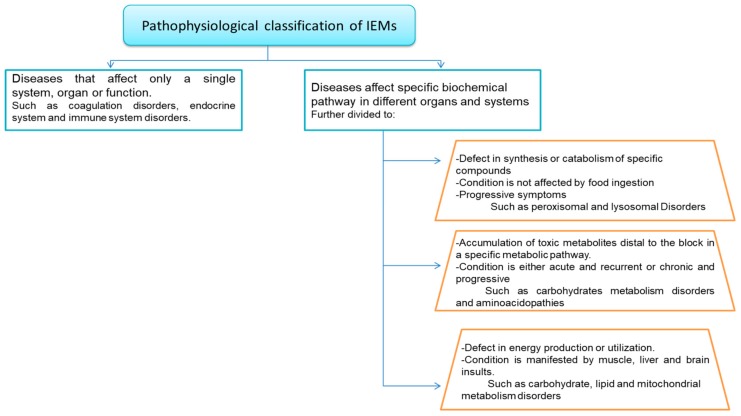
Pathophysiological classification of IEMs adopted from [55].

**Figure 5 metabolites-09-00242-f005:**
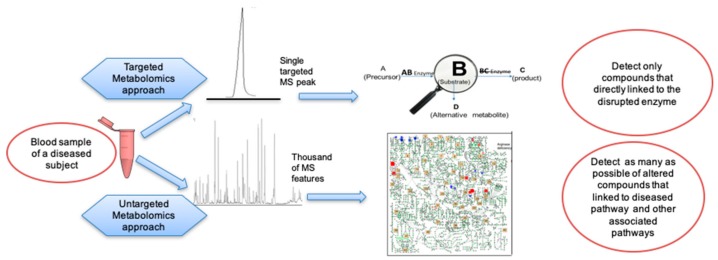
Improvement of IEM diagnosis tests using untargeted metabolomics.

**Figure 6 metabolites-09-00242-f006:**
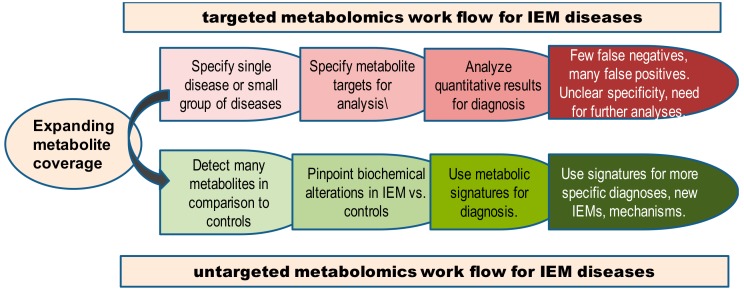
Workflow for diagnosis of IEMs comparing targeted versus untargeted metabolomics.

**Table 1 metabolites-09-00242-t001:** Characteristics of targeted and untargeted metabolomics approaches in IEM diagnosis.

Parameter	Targeted Metabolomics	Untargeted Metabolomics
**Main concept**	Select specific metabolites (10-100) as targets in LC-MS/MS or direct infusion MS/MS to diagnose a specific disease. Detect fragment ions of these metabolic targets and perform molar quantification using internal standards	Detect all ions within a certain mass range in LC-MS/MS and identify as many metabolites as possible. Use signal intensities of both known and unknown metabolites to characterize diseases phenotypes. Quantification can be aided by quality controls, normalizations, and internal standards.
**Instrumentation**	GC-MS (in single ion monitoring) LC-triple quadrupole MS LC-quadrupole linear ion trap MS (in multi-reaction monitoring)	GC-MS (full scan) LC-quadrupole time-of-flight MS LC-orbital ion trap MS
**Weaknesses**	Selective isolation of a group of metabolites. Focus on only specific (target) metabolites may increase the risk of overlooking metabolic responses in other pathways. Target metabolites may lack specificity to classify a variety of IEMs. Costs for internal standards and complexity of data analysis increases with the number of target metabolites.	Maximum number of metabolites. Relative (normalized) signal intensities are not robust inter-laboratory units. Lack of absolute quantification hampers defining ‘normal’ metabolite levels on a population level. Comparisons only based on differentiating groups within studies. Data processing parameters not validated across different software. Compound identification is not standardized yet.
**Strengths**	*Hypothesis testing:* Targeted experiments provide better quantitation, typically by internal standards and specific mass spectrometer conditions. Absolute quantifications of metabolite in may be used to establish baseline metabolite levels for defining healthy versus altered states and for interlaboratory comparison. Identification is performed by comparison to internal standards and specificity of MS/MS.	*Hypothesis generating:* Untargeted experiments provide broader coverage with the potential to screen known compounds and discover novel metabolites. Cover “all” metabolites in samples within the bounds of an analytical technique. Typically >1000 metabolite signals. No increase in the cost when more metabolites are detected. More information about the overall genomic environmental interaction to yield specific IEM phenotypes.

**Table 2 metabolites-09-00242-t002:** Using untargeted metabolomics for identification of inborn errors of metabolism.

Sample	Instrumentation and Platform	Number of Samples	Number of Studied Diseases	Results	ref.
Plasma	LC ESI (−) QTOF C18 column,	24 patients, 21 controls	9 patients with propionic academia, 15 patients with methylmalonic acidemia	Classification by known and new markers	[42]
Dried blood spots	ESI (+,−) Orbitrap Q-Exactive MS	66 patients, 500 controls	9 diseases: PKU, MCADD, HCY, CLD, MSUD, IVA, PA, 3-MCC, Tyrosinemia, citrullinemia galactosemia	Correctly grouped previous false positive cases	[108]
Urine	LC ESI (+) QTOF HILIC amide column	21 patients, 14 controls	4 diseases: cystinuria, maple syrup urine disease, adenylosuccinate lyase deficiency, galactosemia	Groups were correctly classified	[169]
Plasma	GC-MS, ESI (+,−) Orbitrap MS HILIC column	1 patient	Aromatic L-amino acid decarboxylase (AADC) deficiency	Case study	[170]
Plasma	GC-MS, ESI (+,−) LC-MS HILIC column	120 patients 70 controls	21 IEM diseases	20 IEMs classified, novel biomarkers	[168]
Dried blood spots	ESI (+) Orbitrap MS	25 patients 25 controls	Medium Chain Acyl-COA Dehydrogenase Deficiency (MCADD)	Disease groups classified	[171]
Plasma	GC-MS, ESI (+,−) Orbitrap MS HILIC column	4 patients	Adenyl succinate lyase (ADSL) deficiency	Disease characterized	[172]
Plasma	GC-MS, lipidomics by LC-QTOF MS	12 patients, 11 controls	Long-Chain Hydroxy Acyl CoA Dehydrogenase, Carnitine Palmitoyl Transferase 2 Deficiency	Identified with pathway detection	[173]
Urine	LC ESI (+,−) Q-Exactive MS HILIC column	34 patients 66 controls	18 IEM diseases	Characterization	[116]
Skin fibroblasts	LC-ESI (+,−) QTOF MS with HILIC column	3 patients 3 controls	Ethylmalonic Encephalopathy	Detected possible new biomarker	[114]
CSF, urine plasma	GC-MS, LC (+,−) ESI Orbitrap w/ HILIC column	17 patients	Glucose Transporter Type 1 Deficiency Syndrome (GLUT1-DS)	Detected possible new biomarker, pathway affected	[174]
Urine	LC ion mobility MS	49 patients 66 controls	Mucopolysaccharidosis MPS III A, B, C, D	Four phenotypes identified with pathways	[175]
Plasma	LC (+,−) QTOF HILIC column		46 IEM diseases	42 IEM groups, new biomarkers	[118]
Plasma	LC - heated ESI Q-Exactive MS	48 patients	Various types of urea cycle defect (UCD)	Detect novel metabolites, monitor treatment	[176]

Abbreviations; ESI: Electrospray Ionization; LC: Liquid Chromatography; GC: Gas Chromatography; MS: Mass Spectrometry; QTOF: Qudropole time of flight; HILIC: Hydrophilic Interaction Liquid Chromatography; CSF: Cerebrospinal fluid.

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
