# Peer review of "Inborn Errors of Metabolism in the Era of Untargeted Metabolomics and Lipidomics"

_metabolites, 2019, doi:10.3390/metabo9100242_

Round 1

Reviewer 1 Report

Review: Inborn errors of metabolism in the era of untargeted metabolomics and lipidomics

In the review by Ismail et al, the authors have attempted to summarize a great deal of information pertaining to background and general information for those unfamiliar to IEMs as well as give context to IEMs in the era of untargeted metabolomics.  

While this reviewer appreciates the breadth of knowledge attempted to be covered, my general opinion is that this review seems too broad, and as such many of the sections provide for only very broad summaries. Furthermore, it appears that the extent of generalized IEM disease components written is also not necessary for the reader to grasp IEM’s in the era of untargeted metabolomic’s use. For example there is a very heavy focus on summarizing incidence rate as well as new born screening non-technical aspects, before coming to discussions on the key elements of untargeted metabolomics, which start around page 10 with a review of only 14 pages long.

As such, I would suggest to trim the introductory sections and also remove redundancy in statements (see comments below) which would enable a more precise focus and balanced view of IEMs in the era of targeted and untargeted metabolomics. In general I think the authors have set up a perfect outline to do this as summarized at the end of their introduction with the statement to cover IEM disease with 1. emphasis on screening and diagnosis using targeted vs untargeted metabolomics, 2. with matrices used, 3. instrumentation and 4. data processing methods. With finally 5. addressing obstacles to adapt untarged mass spec in IEM screenings.  I would say at present however, this nice outline of information starts only on page 6 (of a 14 page review) and that the section on addressing obstacles is placed throughout, rather than a distinct section which is maybe more desirable.  Preferable to this reviewer and to the readership that I think this review could speak to, would be to de-emphasize the broad introductory components (incidence, SSIEM categorization etc) and expand on reviewing current literature that has used the untargeted and targeted metabolomics strategies and give more specifics of what they show… for example with respect to lowering false discovery rate, impacting costs, patient diagnosis etc.

Specific comments;

1.      Several statements of redundancy and also without references: e.g.  “Most IEMs are inherited in an autosomal recessive (AR) manner “ is written on line 106 and line 99.

2.      Is Figure 2, a summary of the previous literature?  The source should be clearly attached to the figure.

3.      In the abstract the authors make a key statement that this reviewer thinks would be a critical piece to cover “Such targeted approaches have disadvantages such as a higher risk of false negative and false positive”.  As far as this review is aware, investigators have shown that untargeted metabolomics can identify patients and potentially novel biomarkers but they have not (yet) demonstrated that untargeted metabolomics is going to be better than targeted metabolomics with respect to false positive and false negative rates as the authors allude to in the abstract. Please provide adequate perspective here i.e. what have the current studies actually shown (quote real numbers) explain why you expect it will do better and what has to be done to achieve this. Line 261 to 264 is an opportunity to expand on this.

4.      Similarly: in Line 339: Did including the additional Fabry metabolites actually improve false positive and false negative rate? Please expand. Also section 376-391. Please expand and give specifics about how much the false negative and false positive rate improved.

5.      To summarize please be specific about are the current false positive and false negative rate (how big is this problem?), and how much is untargeted metabolomics expected to improve this. The summary in figure 6, shows untargeted pointing to ‘certain’ screening result vs uncertain screening result. I would probably guess there still is a margin of error associated with that and would prefer to have current review of actual numbers supporting this statement.

6.      Discussing untargeted metabolomics as a tool for newborn screening is quite far away from reality as decisions on what to screen for are largely a political / regulatory decision and not so much technical. Statements such as Current newborn screening programs are only performed for treatable diseases. However, availability of treatment options should not be used as an exclusion criterion. Even without available treatments, screening will still have a benefit of raising concerns to nonspecific early symptoms and prenatal diagnosis are reflecting the authors opinion. This is however a very complex issue that warrants much more nuance. Given that this topic is not the primary focus of this review, I recommend removing such statements.

7.      Line 307-308: The statement about tyrosine being used as a biomarker for homocystinuria is incorrect. The primary marker for homocystinuria is methionine. Tyrosine is used for screening of tyrosinemia type 1. Early on it was realized that tyrosine alone was not sufficient (many false positive) and succinylacetone  was added as a second and more specific marker for tyrosinemia type 1. So the statements here are incorrect. Issues with false positives have arisen, but have been dealt with by adding additional metabolites or metabolite ratios as screening targets. Another examples is galactosemia, Measuring galactose alone is known to give many false positives, therefore enzyme activity is measured directly following an elevated galactose screen (is part of the neonatal screen).

Author Response

We would like to thank the editor and the reviewer for their valuable time and we appreciate their helpful comments.

Reviewer 1

Comment: In the review by Ismail et al, the authors have attempted to summarize a great deal of information pertaining to background and general information for those unfamiliar to IEMs as well as give context to IEMs in the era of untargeted metabolomics.

 Reply: Many thanks to the reviewer for this positive feedback, which we appreciate. We agree the review was directed to researchers in the field of untargeted metabolomics together with inborn error of metabolism specialists who interested in new techniques for diagnosis.

Comment: While this reviewer appreciates the breadth of knowledge attempted to be covered, my general opinion is that this review seems too broad, and as such many of the sections provide for only very broad summaries. Furthermore, it appears that the extent of generalized IEM disease components written is also not necessary for the reader to grasp IEM’s in the era of untargeted metabolomic’s use. For example there is a very heavy focus on summarizing incidence rate as well as new born screening non-technical aspects, before coming to discussions on the key elements of untargeted metabolomics, which start around page 10 with a review of only 14 pages long.

Reply: We appreciate the reviewer’s insights. There are a number of general reviews on details of metabolomics technologies, from data acquisition to data processing. In the context of a review on using untargeted metabolomics and IEMs, it is important for readers to understand low incidence rates and the very diverse metabolic effects of IEMs, because especially the latter fact is the reason why untargeted metabolomics needs to complement classic targeted metabolite analyses in IEMs. We revised the manuscript and shortened background information on IEMs now.

Comment: As such, I would suggest to trim the introductory sections and also remove redundancy in statements (see comments below) which would enable a more precise focus and balanced view of IEMs in the era of targeted and untargeted metabolomics. In general I think the authors have set up a perfect outline to do this as summarized at the end of their introduction with the statement to cover IEM disease with 1. Emphasis on screening and diagnosis using targeted vs untargeted metabolomics, 2. with matrices used, 3. Instrumentation and 4. data processing methods. With finally 5. addressing obstacles to adapt untarged mass spec in IEM screenings.  I would say at present however, this nice outline of information starts only on page 6 (of a 14 page review) and that the section on addressing obstacles is placed throughout, rather than a distinct section which is maybe more desirable.  Preferable to this reviewer and to the readership that I think this review could speak to, would be to de-emphasize the broad introductory components (incidence, SSIEM categorization etc) and expand on reviewing current literature that has used the untargeted and targeted metabolomics strategies and give more specifics of what they show… for example with respect to lowering false discovery rate, impacting costs, patient diagnosis etc.

Reply: Thank you for clarifying your point of view on IEM details. Most metabolomic researchers are only vaguely aware of the large diversity of IEMs both in terms of metabolic causes and effects, but also with respect to diversity in analytical procedures and implementations across countries. Clinical biochemists who are responsible in daily routines to perform IEM diagnosis will also benefit from a broader view on the topic. To strengthen the introduction, we have now shortened the classifications section of the review.   and added a section titled ” IEMs screening and diagnosis comparing targeted versus untargeted metabolomics”.                                                                                                                                                                                                                                                 

Specific comments;

Comment: 1. Several statements of redundancy and also without references: e.g.  “Most IEMs are inherited in an autosomal recessive (AR) manner “ is written on line 106 and line 99.

Reply: We have removed the sentence from line 99 and added a reference to the former line 106 (now line 118).  We checked for further redundancy throughout the text.

Comment: 2. Is Figure 2, a summary of the previous literature?  The source should be clearly attached to the figure.

Reply: Correct, figure 2 summarizes results from more than15 references. All references are listed in a supplemental table and is indicated as such.

Comment: 3.      In the abstract the authors make a key statement that this reviewer thinks would be a critical piece to cover “Such targeted approaches have disadvantages such as a higher risk of false negative and false positive”.  As far as this review is aware, investigators have shown that untargeted metabolomics can identify patients and potentially novel biomarkers but they have not (yet) demonstrated that untargeted metabolomics is going to be better than targeted metabolomics with respect to false positive and false negative rates as the authors allude to in the abstract. Please provide adequate perspective here i.e. what have the current studies actually shown (quote real numbers) explain why you expect it will do better and what has to be done to achieve this. Line 261 to 264 is an opportunity to expand on this.

Reply: The very reasoning to write this review is to alert researchers and clinical biochemists of the promise of untargeted metabolomics for newborn screening. Yet, as the reviewer correctly points out, untargeted metabolomics has not been shown to be validated enough in clinical practice as better alternative to targeted screening. We try to make this distinction clear: The diversity of IEMs and the complexity of overlapping metabolic effects demand the use of broad-scale metabolic measurements, instead of targeting only few ‘key’ compounds. That is the promise of untargeted analyses (that still cover known key metabolites!). Preliminary results are presented from a limited number of studies comparing the two different techniques are summarized in the added section” IEMs screening and diagnosis comparing targeted versus untargeted metabolomics”.  Specifically, untargeted metabolomics probes different areas of metabolism that can be combined for diagnostic purposes even if differences to heathy infants are not by orders of magnitude. Secondly, untargeted metabolomics can find unexpected peaks that could later be used in targeted screening. Both areas of research are urgently needed given the wide range of IEMs (and novel IEMs, and overlapping IEMs) that we have laid out in the introduction. Our references point to the current limited efforts in the literature, and we agree with the reviewer that in-depth validations, harmonization and standardization are required to bring untargeted metabolomics to the clinic. We have expanded this argument as suggested by the reviewer.

Comment: 4.      Similarly: in Line 339: Did including the additional Fabry metabolites actually improve false positive and false negative rate? Please expand. Also section 376-391. Please expand and give specifics about how much the false negative and false positive rate improved.

Reply: Thank you for this excellent observation. The same author (Auray-Blais et al.) had performed studies on male and female patients with Fabry disease and reported that the new biomarkers had not been detected in any healthy control urine sample. This indicates the biomarkers (Globotriaosylceramide (Gb3) and its isomers) in their study had no false positive cases and 100% specificity. Currently, female subjects remain underdiagnosed, but with this new biomarker, Fabry disease gets much better diagnosis in female patients. We added this explanation in the new section” IEMs screening and diagnosis comparing targeted versus untargeted metabolomics.

Comment: 5.      To summarize please be specific about are the current false positive and false negative rate (how big is this problem?), and how much is untargeted metabolomics expected to improve this. The summary in figure 6, shows untargeted pointing to ‘certain’ screening result vs uncertain screening result. I would probably guess there still is a margin of error associated with that and would prefer to have current review of actual numbers supporting this statement.

Reply: As discussed above, use of untargeted metabolomics is only now beginning. We now specified points raised by the reviewer, including improved diagnostic outcomes, in the added section IEMs screening and diagnosis comparing targeted versus untargeted metabolomics. As untargeted metabolomics is proposed to complement classic targeted screening, and because it currently awaits improvements in harmonization and standardization, too few papers have exemplified its use to give accurate numbers. We have changed the summary in figure 5 (figure 6 before) to “comprehensive” and “uncertain” wording.

Comment: 6.      Discussing untargeted metabolomics as a tool for newborn screening is quite far away from reality as decisions on what to screen for are largely a political / regulatory decision and not so much technical. Statements such as “Current newborn screening programs are only performed for treatable diseases. However, availability of treatment options should not be used as an exclusion criterion. Even without available treatments, screening will still have a benefit of raising concerns to nonspecific early symptoms and prenatal diagnosis” are reflecting the authors’ opinion. This is however a very complex issue that warrants much more nuance. Given that this topic is not the primary focus of this review, I recommend removing such statements.

Reply: We have clarified that these statements are the authors’ opinions, based on the premise of medicine to provide wider diagnostic options for patient populations. We have added cautionary remarks to give this premise more nuance, for example, cost-benefit ratios of diagnostic tests. Indeed, this sentence was deliberately placed at the end of the review, as it is often used to broaden the horizon and give prospects on future developments.

Comment: 7.      Line 307-308: The statement about tyrosine being used as a biomarker for homocystinuria is incorrect. The primary marker for homocystinuria is methionine. Tyrosine is used for screening of tyrosinemia type 1. Early on it was realized that tyrosine alone was not sufficient (many false positive) and succinylacetone  was added as a second and more specific marker for tyrosinemia type 1. So the statements here are incorrect. Issues with false positives have arisen, but have been dealt with by adding additional metabolites or metabolite ratios as screening targets. Another examples is galactosemia, Measuring galactose alone is known to give many false positives, therefore enzyme activity is measured directly following an elevated galactose screen (is part of the neonatal screen).

Reply: The reviewer is correct, and we have fixed the statements in lines 319-327. For galactosemia, the reviewer is correct that targeting galactose alone for quantitative measurements in neonatal screening gives insufficient evidence for galactosemia. Yet, follow-up analyses (e.g. by enzyme activity analyses) complicate large-scale screenings, add burden to the families and add costs. We turn the argument around, saying that if other metabolites (besides galactose) would be part of the diagnosis (using untargeted metabolomics analyses), fewer follow-up confirmatory tests might be needed.  

Reviewer 2 Report

In their manuscript entitled, “Inborn errors of metabolism in the era of untargeted metabolomics and lipidomics”, the authors review inherited metabolic diseases and metabolomics approaches to diagnosis. This is a timely subject that is of broad general interest and multiple recent publications have discussed this topic (e.g., PMID: 30306077, PMID 27649151, PMID 28675806, PMID: 26712461). Strengths of this manuscript include an in-depth analysis of disease incidence across multiple countries and the comprehensive summarization of untargeted metabolomics studies completed to date in the field of inborn errors of metabolism.  However, in my view this manuscript has a number of limitations in terms of novelty, content and organization that preclude its acceptance for publication at this time.

I would recommend the authors seek out an additional coauthor with expertise in the field of inborn errors of metabolism.  In addition, the paper would benefit from tightening its scope to allow a more comprehensive review of a more focused topic.  Below I have listed a subset of concerns that I have selected to support my conclusion and to help guide the authors to improve this manuscript.

Line 16:  Targeted approaches have a higher risk of false positives compared to what?  I’m not sure I would buy the argument that untargeted metabolic testing would have a lower false positive rate than targeted testing.

Line 44:  I assume screening here is referring to newborn screening.  If so, newborn screening should be explained.  

Line 49:  Please clarify what is meant by different phenotypes of the same disease. Mucopolysaccharidoses II, IVA, and VI are distinct genetic diseases.   To my understanding, the cited paper does not describe phenotypic differentiation within the same disease.     

Line 55:  Do you mean whole genome sequencing and the possibility of incidental findings?

Line 57:  Newborn screening is mentioned for the first time here (outside of the abstract).  I believe you are alluding to newborn screening in much of your preceding introduction.  It would be good to discuss NBS sooner and in more detail.  

Line 129:  typo, assort disorders? 

Line 240:  typo, generate of

Line 319:  Standardized sets of basic newborn screening tests are lacking.”  Multiple professional groups have made recommendations regarding appropriate newborn screening testing.  ACHDNC recommended uniform screening panel,  ACMG PMID 16783161 

Line 324-25:  “Current newborn screening programs are only performed for treatable diseases. However, availability of treatment options should not be used as an exclusion criterion.”  I strongly disagree with this statement.   The Wilson and Jungner criteria for population screening should be reviewed and the authors should explain why they disagree.  

Line 378-380:  "Untargeted metabolomics may overcome problems in targeted methods for newborn screening by increasing the number of screened IEM diseases and decreasing the incidence of false negative results [102]."  

It’s unclear to me what connects this statement and this citation.   The cited paper broadly reviews newborn screening.  It does not support the assertion that untargeted metabolomics may overcome issues with NBS. 

On this topic, false negatives appear to be a very minor problem in NBS (see a different paper from the authors of [102] which discusses false negatives by NBS, PMID 24970580). The authors should also discuss the limitations of untargeted metabolomics in newborn screening, e.g., cost, analysis time, and incidental findings.      

General comments

Introduction:  I think the introduction could use quite a bit of work in terms of structure and content. In some cases the introduction goes into too much detail on tangential points while in other cases it fails to provide sufficient explanation of key concepts.   For example, the introduction spends quite a bit of time discussing the history of now irrelevant technologies used for detecting inborn errors of metabolism but never defines targeted vs untargeted metabolomics.  Similarly, I would suggest providing a more complete description of newborn screening and at an earlier point in the introduction.  

Untargeted metabolomics:  Urine organic acid screening by GC-MS could be considered the first untargeted metabolomics test used to diagnose inborn errors of metabolism and it is still in routine use today.  It would be good to mention this technique in your discussion of untargeted methods.

Figures and tables 

Box1-  This adds little to the paper and could be removed. If you choose to keep this figure I would reorganize the content and fix the typos/grammatical errors. 

Figure 4-  organization of boxes is confusing and unnecessary.   This could be a table or removed

Figure 5-  This would probably be better presented using a table.  

Table1-  Many of the entries listed here are not absolutely true and/or are confusing, (untargeted results are not always normalized in the way described,  internal standards are still used in some untargeted applications to broadly control for loss during sample prep, the use of hypothesis seems out of place here, etc).  It might be better to condense the number of rows and present this more as the strengths and weaknesses of each approach.  

Author Response

We would like to thank the editor and the reviewer for their valuable time and we appreciate their helpful comments.

Reviewer 2

Comment: In their manuscript entitled, “Inborn errors of metabolism in the era of untargeted metabolomics and lipidomics”, the authors review inherited metabolic diseases and metabolomics approaches to diagnosis. This is a timely subject that is of broad general interest and multiple recent publications have discussed this topic (e.g., PMID: 30306077, PMID 27649151, PMID 28675806, PMID: 26712461). Strengths of this manuscript include an in-depth analysis of disease incidence across multiple countries and the comprehensive summarization of untargeted metabolomics studies completed to date in the field of inborn errors of metabolism.  However, in my view this manuscript has a number of limitations in terms of novelty, content and organization that preclude its acceptance for publication at this time.

Reply: Many thanks to the reviewer for this positive feedback, which we appreciate. We have used  your suggestions to improve our paper contents and organizations. All four reviews focused on the interaction of metabolite data and genomic variants, whereas our review focuses on techniques and example studies, highlighting how untargeted metabolomics may be better suited to fulfill the need for specificity in IEM diagnosis, as well as extending the breadth of metabolic data to characterized currently undiagnosed IEMs. Specifically, PMID: 30306077 (Mussap et al., 2018) was already cited in our review before. It discusses a query of the human metabolome data base (HMDB), stating that (based on literature behind HMDB) 247 metabolites are associated with 147 IEMs and 202 metabolic pathways. No target analysis would be able to perform quantifications of absolute concentrations of this number of metabolites involved in various IEMs. PMID 28675806 (Sandlers, 2017) was also already cited in our review. Sandlers focused on details of selected studies and initiatives to improve coverage, reporting and re-use of metabolomics data, but this review did not exemplify how metabolomics could be used to improve classification of IEM categories. PMID: 26712461 (Argmann et al., 2016) and PMID 27654911 (Tebani et al., 2016) both focused on integration of multiple omics data sets, with very little emphasis on metabolomics. We have now cited these reviews to establish a comprehensive record, but there is little overlap with the direction or focus of the review presented here. Yet, all four reviewers emphasize the need to extend analytical capabilities beyond the current state of IEM diagnosis.

Comment: I would recommend the authors seek out an additional coauthor with expertise in the field of inborn errors of metabolism.  In addition, the paper would benefit from tightening its scope to allow a more comprehensive review of a more focused topic.  Below I have listed a subset of concerns that I have selected to support my conclusion and to help guide the authors to improve this manuscript.

Reply: Thank you for your suggestion to further strengthen the review by focusing on a specific topic (i.e. a specific IEM). The argument to use untargeted metabolomics is that it will allow to broaden the view and scope of diagnosis of different IEMs, including metabolic patterns that overlap between different IEMs. It is entirely difficult to tightening this point of view, especially because untargeted metabolomics is still new to the field of IEM research and because untargeted metabolomics has been exemplified in different IEM diseases such as Fabry’s disease. We therefore added a section “IEMs screening and diagnosis comparing targeted versus untargeted metabolomics” where we took Fabry’s disease as a specific example. With respect to adding another author, we would like to state that first author Ismail is actively using clinical biochemistry to diagnose IEMs on a daily basis in her work in clinical practice in Egypt, with many subjects presenting with unclear phenotypes of complex origins. Hence, expertise in IEMs and the focus on extending the basis of IEM diagnosis towards larger metabolic panels (and using untargeted metabolomics) is not based on assumptions but founded on practical observations in the clinic.  

Comment: Line 16:  Targeted approaches have a higher risk of false positives compared to what?  I’m not sure I would buy the argument that untargeted metabolic testing would have a lower false positive rate than targeted testing.

Reply: Some IEM diseases such as PKU are known to have very clear and straightforward diagnostic criteria, and indeed, then there is little concern for false positive diagnoses. Other (more complex) IEMs, including galactosemia, require additional data for true positive diagnosis, as pointed out by the reviewer. Yet, the reviewer is correct by arguing that untargeted metabolomics has not been proven to yield lower false positive IEM detections than targeted approaches, mainly because it is still an early and developing method.  We agree with the reviewer and have removed that remark.

Comment: Line 44:  I assume screening here is referring to newborn screening.  If so, newborn screening should be explained. 

Reply: We thank the reviewer for this helpful comment. We have now added a new paragraph on page 1, lines 36-42, in which we explain newborn screening in brief. In the section on IEM diagnosis, this explanation is extended on page 6, lines 191-203

Comment: Line 49:  Please clarify what is meant by different phenotypes of the same disease. Mucopolysaccharidoses II, IVA, and VI are distinct genetic diseases.   To my understanding, the cited paper does not describe phenotypic differentiation within the same disease.

 Reply: We thank the reviewer for this helpful comment. We have now replaced it (page 2, lines 54)

Comment: Line 55:  Do you mean whole genome sequencing and the possibility of incidental findings?

Reply: Sorry for the unclear wording. Whole genome sequencing is usually too expensive to be performed on large scale. We refer to NGS genotyping. (Page 2, lines 61-64)

Comment: Line 57:  Newborn screening is mentioned for the first time here (outside of the abstract).  I believe you are alluding to newborn screening in much of your preceding introduction.  It would be good to discuss NBS sooner and in more detail.  

Reply: We thank the reviewer for this helpful comment. We have now added a new paragraph on page 1, lines 36-42 which we explain newborn screening in brief. In the section on IEM diagnosis, this explanation is extended page 6, lines 191-203

Comment: Line 129:  typo, assort disorders? 

 Reply: Corrected to “Classic classification of IEMs were assorted as disorders of carbohydrate metabolism” in page 4, lines 140-141

Comment: Line 240:  typo, generate of

Reply: Corrected to “matrix to generate metabolic signatures” in page 7, lines 254

Comment: Line 319:  “Standardized sets of basic newborn screening tests are lacking.”  Multiple professional groups have made recommendations regarding appropriate newborn screening testing.  ACHDNC recommended uniform screening panel,  ACMG PMID 16783161 

Reply:  We have clarified our comment in page 9, lines 335-339 as ” The American College of Medical Genetics have recommended a uniform panel of conditions formed of 29 diseases together with 25 related diseases, including outcomes and guidelines [122]. However, these recommendations are given for a specific country but are not used worldwide. Even within the U.S., different states adopted varying parts of these recommendations and did not implement single standardized sets [123].” This is the very reason why we wrote a larger introductory section and broadened the scope of this review.

Comment: Line 324-25:  “Current newborn screening programs are only performed for treatable diseases. However, availability of treatment options should not be used as an exclusion criterion.”  I strongly disagree with this statement.   The Wilson and Jungner criteria for population screening should be reviewed and the authors should explain why they disagree.  

Reply:  We respect the strong opinion of the reviewer, and indeed, there are risks for over-diagnosis of diseases in populations as well as for under-diagnosis. In many complex diseases (such as Alzheimer’s), there is no current treatment but a broad consensus in the society and academia that we need diagnostic tools and need to learn more about the origins, scope, manifestations and the onset and progression of the disease, all with the ultimate aim to find possible therapeutic options. For IEMs, we need to recognize that there are many patients who present at the clinic with ambiguous manifestations that lack clear diagnosis with current tests. Parents of these patients would be comforted to some extent if at least the symptoms and possibly metabolic pathways and underlying genes would be pinpointed and named, even if there is no current treatment. In the United States, the NIH even funds initiatives to diagnose and screen and monitor through the “Undiagnosed disease network”.

With respect to the classic Wilson and Jungner criteria, the WHO remarked the uneven adoption of the principles in the era of genomic testing (PMID: 18438522 ),in (PMID: 16763907) citing  Pollitt  “The lack of even broad concordance at the level of national policy is extremely disturbing. Though all discussion is nominally founded on the ten principles laid down by Wilson and Jungner in 1968, there seems no generally accepted way of using these principles, or derived criteria, as objective decision tools.” Moreover, these original 10 principles have now been expanded to 41 sets and 367 unique principles in a meta-analysis of using principles to decide which diagnostic tools to be used on a population screening level (PMID: 29632037)

We have added nuances and clarifications to our statement, lines 344-361 page9&10. We have an equally strong opinion, yet opposite view to the reviewer. We think the time has come to call for expanded metabolite screening and improved diagnosis based on metabolite patterns (and major aberrations in unexpected metabolites) that utilized novel technologies, following the path laid out by genetics and genomics over the past 20 years. We think IEM patients are underserved by current assays, and we see similar movements in other areas of clinical research that aim to understand complex diseases from a metabolic perspective (Alzheimer’s, Myalgic / Chronic Fatigue Syndrome, Autism and others), all of which lack current treatment options. We cite a community White Paper on this trend towards using metabolomics in Precision Medicine to support the notion that extended metabolic assays can not only lead to better diagnosis but also to improve understanding of underlying etiologies, including for currently undiagnosed diseases.

Comment: Line 378-380:  "Untargeted metabolomics may overcome problems in targeted methods for newborn screening by increasing the number of screened IEM diseases and decreasing the incidence of false negative results [102]."  

It’s unclear to me what connects this statement and this citation.   The cited paper broadly reviews newborn screening.  It does not support the assertion that untargeted metabolomics may overcome issues with NBS. 

Reply: Thank you for checking! We now cite the correct reference by the same author “Expanded newborn screening in New South Wales: missed case”. We would like to state that the sentence reads “may overcome problems”, and indeed, we clarify that the technology has not yet evidenced broad-scale validations of this premise.

Comment: On this topic, false negatives appear to be a very minor problem in NBS (see a different paper from the authors of [102] which discusses false negatives by NBS, PMID 24970580). The authors should also discuss the limitations of untargeted metabolomics in newborn screening, e.g., cost, analysis time, and incidental findings. 

Reply: The reviewer is correct that thresholds and cut-off values for NBS are regularly set at low concentrations to avoid false negative detections, while logically expanding the number of false positive detections. We have cited this reference now (Estrella 2014). We also added a specific section on current limitations of untargeted metabolomics, specifically with respect to quantitative aspects and harmonization and standardization of assays. To achieve this aim, the scientific community will need authoritative databases that give ranges of normal (‘healthy’) subjects when screened by untargeted metabolomics. Only such databases would enable us to define thresholds (such as 6-sigma) beyond which we would be able to state truly abnormal metabolic findings. Costs will relate to current assays: sample preparation and staff time are roughly similar in our own metabolomic centers when using targeted assays (for steroids, oxylipins, bile acids and similar low abundant metabolites) or using untargeted assays (for complex lipids, biogenic amines and primary metabolites). Analysis time is shorter in untargeted assays when broken down by the number of analytes: our biogenic amines assay, for example, typically detects about 500 identified metabolites in two 15 minute runs, while targeted assays (e.g. for 40 steroids) are performed in a single 15 minute run. Per analyte, untargeted analyses are faster. In our center, using Amazon cloud services, costs are $0.20 per sample for data processing with up to 5,000 files processed per day. We agree that not all metabolomic centers have this capacity (yet), but it is definitely at the same speed as targeted assays. With respect to machine costs (investments), accurate mass spectrometers and UPLCs are about 3x more expensive than current triple quadrupole MS instruments, hence, these costs are indeed higher. Yet, such costs are relative to the use per year: if depreciation and maintenance costs are about $100,000 per year, with at least 10,000 analyses run per year and instrument (which we routinely show in our Center over the past 15 years), instrumentation costs for untargeted metabolomics is at about $10 per sample, compared to ~ $3 per sample in targeted assays. For the U.S., such costs are bearable, while in Egypt, even the costs of $6 per sample for deuterated internal standards is too high (covering acylcarnitines and amino acids only, current costs at first author Ismail’s facility).

These arguments are obviously too long to be written in the review. Yet, untargeted metabolomics is not necessarily slower or prohibitively expensive, yet provides larger views on metabolism. At least at UC Davis, it is now mature enough to fill databases and migrate from the academic lab to the clinic, which we currently do by adding facilities at the UC Davis medical center in Sacramento. We have added short comments on these prospects and limitations.

General comments

Comment: Introduction:  I think the introduction could use quite a bit of work in terms of structure and content. In some cases the introduction goes into too much detail on tangential points while in other cases it fails to provide sufficient explanation of key concepts.   For example, the introduction spends quite a bit of time discussing the history of now irrelevant technologies used for detecting inborn errors of metabolism but never defines targeted vs untargeted metabolomics.  Similarly, I would suggest providing a more complete description of newborn screening and at an earlier point in the introduction

Reply: We have altered the structure of the introduction accordingly and added a paragraph on NBS. There are excellent reviews on metabolomic techniques; it would go too far to include differences and nuances for both targeted and untargeted methods. Table 1 distinguishes both approaches.

Comment: Untargeted metabolomics:  Urine organic acid screening by GC-MS could be considered the first untargeted metabolomics test used to diagnose inborn errors of metabolism and it is still in routine use today.  It would be good to mention this technique in your discussion of untargeted methods.

Reply:  Correct, we have added a reference and comment.

Figures and tables 

Comment: Box1- This adds little to the paper and could be removed. If you choose to keep this figure I would reorganize the content and fix the typos/grammatical errors

Reply:  We now removed it and replaced it with the figure in page 5

Comment: Figure 4-  organization of boxes is confusing and unnecessary.   This could be a table or removed

Reply: the figure is removed now

Comment: Figure 5-  This would probably be better presented using a table.

Reply:  We deleted figure 5 as it is represented sufficiently in the text.

Comment: Table1-  Many of the entries listed here are not absolutely true and/or are confusing, (untargeted results are not always normalized in the way described,  internal standards are still used in some untargeted applications to broadly control for loss during sample prep, the use of hypothesis seems out of place here, etc).  It might be better to condense the number of rows and present this more as the strengths and weaknesses of each approach. 

Reply:  We removed some rows and focus on weaknesses and strengths of both approaches

Reviewer 3 Report

Ismail and colleagues review the literature on Inborn Errors of Metabolism (IEMs) and the advantages/disadvantages of targeted vs. untargeted metabolomics for their diagnosis. The Authors make the point that untargeted metabolomics will not only be a useful tool for diagnosing IEMs, especially in Newborn Screening Programs, but also a mean for accessing hitherto unknown alterations in metabolism.

I find this review very informative, well focused, argued, written and referenced. I have no further suggestions of improvement.

Author Response

Reviewer 3 stated:

I find this review very informative, well focused, argued, written and referenced. I have no further suggestions of improvement.

Author comments:

we thank you for reading the paper.

Round 2

Reviewer 1 Report

I think the authors have addressed the majority of this reviewers concerns. I would suggest some careful checking of the grammar/spelling in certain places including the tables, as this reviewer noted some errors.

Author Response

We appreciate the editor’s efforts with respect to our manuscript. We have addressed the issues raised by the editor in the following way:

Comment: Thank you for the edited revision to your paper - whilst there are significant improvements in this version, which are welcomed, it is the view of the editor that the manuscript is still quite a long way from being ready for publication. This revision appears not to have been proof-read thoroughly prior to re-submission as there are numerous small 'tracked-change' type errors including many spaces, missing full stops and a lack of appropriate punctuation etc. Examples are provided in the attached but the whole manuscript needs carefully proofreading. There are also quite a number of sentence structure issues, repetition and lack of clarity in places. The referencing is much better but still needs attention in places. Specific examples in the first 300 lines are indicated in the attachment but it is not exhaustive. 

Reply: We hope we have now performed better proofreading of the manuscript. We have now addressed appropriate punctuations throughout the manuscript.  

General comment: Current state of manuscript remains poorly proof-read. There are many incidences of extra spaces, lost spaces, gaps between full stops etc (e.g. line 114, 120 etc). Other sentences make no literal sense (e.g. line 140 as written). English is poorly structured in many places. The manuscript requires thorough proof reading and input from all authors.

Reply: We regret there were problems with the sentences structure and punctuations. The paper has been carefully revised by to improve punctuations and readability. We fixed the full stop in line 114 (now line 125) and extra space in line 120 (now line 130). The sentence in line 150 (now line 150) has been rephrased to give the intended meaning.

Specific examples (not exhaustive, not read beyond line 300)

Reply: We have now corrected the errors and punctuations through all the manuscript.

Line 37-38: not clear what is meant- re-phrase – refs?

Reply: We have adjusted the text to be clearer. Sentences were added in lines from 38-40, including references.

Line 40: “especially through the advances in genomics [5].” Rephrasing needed/not clear how genomics is implicated in uneven adoption. Also context not clear – which parts of the world?

Reply: We are sorry that the sentence in line 40 was truncated. We now rephrase it to give the intended meaning (now in lines 64-65).

Line 37-42: all via GC-MS, why was uptake slow? some comment on state of analytical capabilityies (pros and cons) if LC- MS/MS then makes a significant positive difference (line 43)…

Reply: In lines 42-50, we now give a brief comparison of these techniques. The main comparison, however, remains in the section ‘Metabolomics technologies used in inborn errors of metabolism’ of the manuscript.

Line 43-44: odd to summarise about mass spec only here as the discussion has been on GC-MS and LC-MS?

Reply: GC-MS and LC-MS have been added to the summary in lines 42-50.

Line 52-54: re-phrasing needed – repeating works (eg ‘more’ meaning not clear in places.
Reply:  We now rephrase and improved the clarity of the sentence.

Line 59 and 60: references needed

Reply: More references were added to paragraph in lines 72-78.

Line 59-61: Heterogeneity in disease or NBS programs? –unclear

Reply: Various NBS programs showed heterogeneity in the diseases that were screened. NBS programs screen a different number of diseases, but also different types of diseases. NBS programs differ even in the cutoff point used to diagnosis. More examples are now added in lines 72-78 to make the point clearer to the reader.

Line 78: “apex of omics trilogy” what does this mean?

Reply: The sentence is now replaced.

Line 114: full stop/capital letters

Reply: We added full stop.

Line 156: Clearly not at ‘any time’ in life then (as implied) considering the next sentence!

Reply:  We have changed to ‘various ages’ to improve the clarity of this sentence, matching the rest of paragraph in line 167.

Line 165-173: Lack of detail throughout exemplified in this section: discussion of symptoms would be enhanced with exmaples of IEMs they relate to. Line 171-173 – some examples? What IEMS?

Reply: More details about the clinical presentations of IEMs were added. We now give examples of diseases categories related to those symptoms in lines 186-206.

Line 176 –eg? +refs?

Reply: Examples for the diseases related to specific manifestations and reference were added in lines 202-207

Line 183-184: re-phrase – unnecessary repeated use of ‘diagnosis’

Reply:  The sentence was rephrased.

Line 186: “abnormal metabolic biomarkers” it’s the levels of normal metabolites that are abnormal in majority of cases – re-phrase.

Reply: The sentence was rephrased.

Line 197: NBS already define above

Reply:  The abbreviation is now used.

Line 214: title is ambiguous. This is basically the use of MS and NMR techniques in NBS – There relevance of ‘metabolomics technologies’ is not made clear.

Reply: It has been changed to ‘Metabolomics technologies used in inborn errors of metabolism’ in line 237.

Line 247: “Recent advances in HRMS” – this is cryptic and unclear what does ‘recent’ mean? For those who know still not clear whether your referring to orbitrap technology, FT-ICR-MS, or Q-TOF technology or some combination?

Reply: In line 270, HRMS was referred to QTOF MS, TTOF MS and quadrupole Orbitrap MS instruments. Ion cyclotron resonance MS is not used in IEM screening. HRMS was further elaborated in the section ‘Untargeted metabolomics in the screening and diagnosis of inborn errors of metabolism’.

Line 291-292: Seems this is an important statement – use of “might” is ambiguous – has it been shown that untargeted can provide same level of ‘quantification confidence’ in an example or not? – would help to have references. Its also not entirely clear what is meant by “quantification confidence” (LOD, sensitivity, precision, accuracy? Some combination?

Reply: There is no principle difference between one-point calibrations by internal standards in ‘targeted’ or ‘untargeted’ assays. The cited reference is a review paper on many published papers in untargeted metabolomics and lipidomics, but it does not cover IEM cases. In lipidomics, use of internal standards is very common, but often applied not only to one specific compound (like in targeted analyses) but to a group of compounds (such as ‘all phosphatidylcholines’). The accuracy of this approach has not been shown in detail, therefore we remain with the simple wording “might”. To our knowledge, this principle has not yet been used in untargeted analyses of IEM diseases (hence ‘might’), but for exudative pleural effusions, the principle has been shown to be very effective (https://doi.Org/10.1016/j.cca.2017.12.003) .

Reviewer 2 Report

A primary concern expressed in my initial review had to due with the ambitious scope of this manuscript and the author’s, at times, confusing and/or misleading review of topics related to inborn errors of metabolism.  The coauthors of this manuscript clearly have strong understanding of the technical aspects of metabolomics.  It is my feeling that this manuscript would greatly benefit from narrowing the scope to this topic.  

In response to my comments, the authors expanded the manuscript.   Comments were added or retained that I feel are confusing or misrepresentations of the state of biochemical genetics testing (a subset of examples are listed below).  Overall, it is my feeling that the response did not adequately address my concerns. 

Line 54 -  now changed to “different genotypes of the same disease”  

To my understanding this paper describes a method to detect different diseases- mucopolysachardoses  types 2, 4A, and 6.  These three diseases are caused by defects in three different genes and so it is not appropriate to call them the same disease.  The paper does not provide any insight on metabolite differences between different genotypes within the same disease.   

Line 78- On the other hand, untargeted metabolomics was introduced at the apex of omics trilogy(proteomics and genomics)[27].

I’m not sure what is meant by this statement? 

Line 122:  Homozygote gene affection can occur from either consanguineous marriage or by a mutation in the other allele to produce the same metabolic effect

What about non consanguineous marriage of two carriers of the same variant?  

Line 188:   all biochemical diagnostics procedures rely on the identification of abnormally high levels of the main substrate.

What about enzyme assays?

Author Response

(The authors gave the same response as above.)
